# HAYSTAC: A Bayesian framework for robust and rapid species identification in high-throughput sequencing data

**Evangelos A. Dimopoulos**[1,☯]*, **Alberto Carmagnini**[2,3☯], **Irina M. Velsko**[4],
**Christina Warinner**[4,5], **Greger Larson**[1], **Laurent A. F. Frantz**[2,3‡]*, **Evan K. Irving-Pease**[1,6‡]*

**1** The Palaeogenomics and Bio-archaeology Research Network, Research Laboratory for Archaeology and History of Art, University of Oxford, Oxford, United Kingdom, **2** Palaeogenomics Group, Institute of Palaeoanatomy, Domestication Research and the History of Veterinary Medicine, Ludwig Maximilian University, Munich, Germany, **3** School of Biological and Behavioural Sciences, Queen Mary University of London, London, United Kingdom, **4** Department of Archaeogenetics, Max Planck Institute for Evolutionary Anthropology, Jena, Germany, **5** Department of Anthropology, Harvard University, Cambridge, United States of America, **6** Lundbeck Foundation GeoGenetics Centre, GLOBE Institute, University of Copenhagen, Copenhagen, Denmark

☯ These authors contributed equally to this work.
‡ These authors co-supervised this work.
* ea.dimopoulos@gmail.com (EAD); laurent.frantz@lmu.de (LAFF); evan.irvingpease@gmail.com (EKL-P)

**Data Availability Statement:** HAYSTAC is available from https://github.com/antonisdim/HAYSTAC and the code to reproduce the analyses shown in the

## Abstract

Identification of specific species in metagenomic samples is critical for several key applications, yet many tools available require large computational power and are often prone to false positive identifications. Here we describe High-AccuracY and Scalable Taxonomic Assignment of MetagenomiC data (HAYSTAC), which can estimate the probability that a specific taxon is present in a metagenome. HAYSTAC provides a user-friendly tool to construct databases, based on publicly available genomes, that are used for competitive read mapping. It then uses a novel Bayesian framework to infer the abundance and statistical support for each species identification and provide per-read species classification. Unlike other methods, HAYSTAC is specifically designed to efficiently handle both ancient and modern DNA data, as well as incomplete reference databases, making it possible to run highly accurate hypothesis-driven analyses (*i.e.*, assessing the presence of a specific species) on variably sized reference databases while dramatically improving processing speeds. We tested the performance and accuracy of HAYSTAC using simulated Illumina libraries, both with and without ancient DNA damage, and compared the results to other currently available methods (*i.e.*, Kraken2/Bracken, KrakenUniq, MALT/HOPS, and Sigma). HAYSTAC identified fewer false positives than both Kraken2/Bracken, KrakenUniq and MALT in all simulations, and fewer than Sigma in simulations of ancient data. It uses less memory than Kraken2/Bracken, KrakenUniq as well as MALT both during database construction and sample analysis. Lastly, we used HAYSTAC to search for specific pathogens in two published ancient metagenomic datasets, demonstrating how it can be applied to empirical datasets. HAYSTAC is available from **https://github.com/antonisdim/HAYSTAC**.

paper is available from https://github.com/antonisdim/haystac_paper.

**Funding:** E.A.D. was supported by the DTP in Environmental Research and funded by NERC award NE/L002612/1. L.A.F.F. and G.L. were supported either by a European Research Council grants (ERC-2013-StG-337574-UNDEAD and ERC-2019StG-853272-PALAEOFARM) and Natural Environmental Research Council grants (NE/K005243/1, NE/K003259/1, NE/S007067/1, and NE/S00078X/1). E.A.D, L.A.F.F. and A.C. were also supported by the Wellcome Trust (210119/Z/18/Z). The funders had no role in study design, data collection and analysis, decision to publish, or preparation of the manuscript.

**Competing interests:** The authors have declared that no competing interests exist.

## Author summary

The emerging field of paleo-metagenomics (i.e., metagenomics from ancient DNA) holds great promise for novel discoveries in fields as diverse as pathogen evolution and paleoenvironmental reconstruction. However, there is presently a lack of computational methods for species identification from microbial communities in both degraded and nondegraded DNA material. Here, we present "HAYSTAC", a user-friendly software package that implements a novel probabilistic model for species identification in metagenomic data obtained from both degraded and non-degraded DNA material. Through extensive benchmarking, we show that HAYSTAC can be used for accurately profiling the community composition, as well as for direct hypothesis testing for the presence of extremely low-abundance taxa, in complex metagenomic samples. After analysing simulated and publicly available datasets, HAYSTAC consistently produced the lowest number of false positive identifications during taxonomic profiling, produced robust results when databases of restricted size were used, and showed increased sensitivity for pathogen detection compared to other specialist methods. The newly proposed probabilistic model and software employed by HAYSTAC can have a substantial impact on the robust and rapid pathogen discovery in degraded/shallow sequenced metagenomic samples while optimising the use of computational resources.

This is a *PLOS Computational Biology* Methods paper.

## Introduction

Metagenomics allows high-throughput sequencing of complex microbial communities that may not be possible to grow in a laboratory. Accurate identification of specific species within a complex community is critical for metagenomic applications in a wide variety of fields, including medicine, microbiome sciences, environmental sciences, biosurveillance, and biomolecular archaeology. Confidently assessing the presence of individual species is also crucial for taxonomic profiling [1], inferring genomic evolution and phylogenetics [2], tracing disease outbreaks [3], generating diagnostics from biofluids [4] and studying ancient pathogens [5].

Recent studies of ancient pathogens based on high-throughput sequencing technology have provided unique insights into the evolutionary history of major human pathogens. These include the etiologic agents of both the Justinian and Black Death plagues (*Yersinia pestis*; [6,7,8,9]), tuberculosis (the *Mycobacterium tuberculosis* complex; [10,11]), syphilis (*Treponema pallidum*; [12,13]) and smallpox (*Variola major* and *Variola minor*; [14]).

Accurately identifying specific microbial species from ancient metagenomes, however, can be challenging for numerous reasons. Firstly, short read sequences originating from closely related species may be difficult to distinguish from each other. In addition, the species of interest may have low abundance in the sample, or the genomes in the reference database may not be representative of the strain present in a sample [15]. Contamination from laboratory environment or reagents may also obscure identification [16]. Identifying specific species within ancient metagenomes is further complicated by issues inherent to ancient DNA, including short molecules (typically between 30–60 bp [17]), post-mortem miscoding lesions (e.g., cytosine deamination), contamination from soil and sediment bacteria in the burial environment, sample handling, and storage [18].

When present in ancient metagenomes, pathogen DNA is usually found in low abundance [15], and researchers often initially screen many samples before deciding which libraries should be more deeply sequenced in order to obtain whole genome data, or be further processed using targeted capture techniques [19,20,5]. Screening for pathogens is typically performed by shallow shotgun sequencing followed by *in silico* analyses of short read data using taxonomic classifiers. Following screening, libraries with a positive identification for a species of interest are then more deeply sequenced and investigated, with or without targeted capture [5]. Confident species identification from metagenomic screening data is therefore critical for studies of ancient pathogens.

Tools for detecting microbial species from metagenomes include metagenomic classifiers that report taxonomic relative abundances [21], such as Kraken2 [22,23], KrakenUniq [24] and the Megan Alignment Tool (MALT) [25], as well as species identification pipelines that classify individual reads, such as Sigma [3], Strain Prediction and Analysis using Representative Sequences (SPARSE) [26], Bayesian Reestimation of Abundance with KrakEN (Bracken) [27], and Heuristic Operations for Pathogen Screening (HOPS, specifically designed for ancient DNA) [28]. The metagenomic classifiers Kraken2, KrakenUniq and MALT provide a good starting point for exploratory studies, especially when it is not known which pathogens may be present [29]. These tools, however, can be slow, require a large amount of RAM [24,25,28], and are prone to false positive identifications of low abundance species [30]. The species identification pipelines Sigma [3] and SPARSE [26] are based on more sophisticated statistical models but they require alignment to a large database, which can be slow [26], and do not provide statistical support for individual species identifications.

To address these issues, we developed High-AccuracY and Scalable Taxonomic Assignment of MetagenomiC data (HAYSTAC - https://github.com/antonisdim/HAYSTAC), a lightweight, fast, and user-friendly species identification tool. HAYSTAC evaluates the presence of a particular species of interest in a metagenomic sample and provides statistical support for the species assignment. Our method is designed to estimate the probability that a specific taxon is present in a metagenomic sample given a set of sequencing reads and a database of reference genomes. We first derive the probability that the set of reads has originated from a single species (single source identification). This is useful, for example, for taxonomic identification of unknown source material (e.g. [31]). We then expand this method to identify multiple species in a metagenomic sample (multiple source identification).

To apply HAYSTAC, the user begins by constructing a database from publicly available genomes via a user-friendly automated process. Sample reads are then mapped to all reference genomes in the database using a Bowtie2 wrapper [32]. HAYSTAC then applies a novel Bayesian framework to infer abundance and statistical support for each species identification, and provides a per-read classification, allowing for both hypothesis-driven and exploratory analyses. HAYSTAC can build and make use of arbitrarily sized databases, with minimal effects on sensitivity and specificity. This means that HAYSTAC can provide reliable species identifications by aligning reads to a handful of reference genomes (instead of thousands), without misidentifying taxa because of the restricted database size. This feature dramatically improves its performance relative to other methods. Altogether, HAYSTAC provides a robust method to assess the presence of a specific species in a metagenomic sample without the need to run multiple programs to assess assignment accuracy, such as is required by other methods (*i.e.*, Bracken for Kraken2 [27] and HOPS for MALT).

To evaluate HAYSTAC's performance, we applied it to both simulated microbiome datasets and to empirical datasets that were generated from DNA extracts derived from archaeological human remains. We further used HAYSTAC to assess the presence of putative respiratory pathogens in oral microbiome-derived metagenomes from ancient human dental calculus.

While dental plaque is known to harbour respiratory pathogens [33], few studies to date have reported these species in its mineralized analogue, dental calculus [34,35], a discrepancy that may reflect identification biases in studies of the latter. Improved identification of ancient respiratory pathobionts (host-associated microbial taxa that may cause disease under specific conditions) has the potential to add to our understanding of pathogen evolutionary dynamics and even aid vaccine development.

## Results

### Method overview

HAYSTAC is a metagenomic species identifier that works with arbitrary user defined databases. The database can be built from any combination of an NCBI search query, a user specified list of NCBI accession codes, the RefSeq representative database of prokaryotic species, or a list of user provided reference assemblies. To construct the database, HAYSTAC uses a heuristic method to select a single representative genome per taxon from those available in the user provided input data (Fig 1).

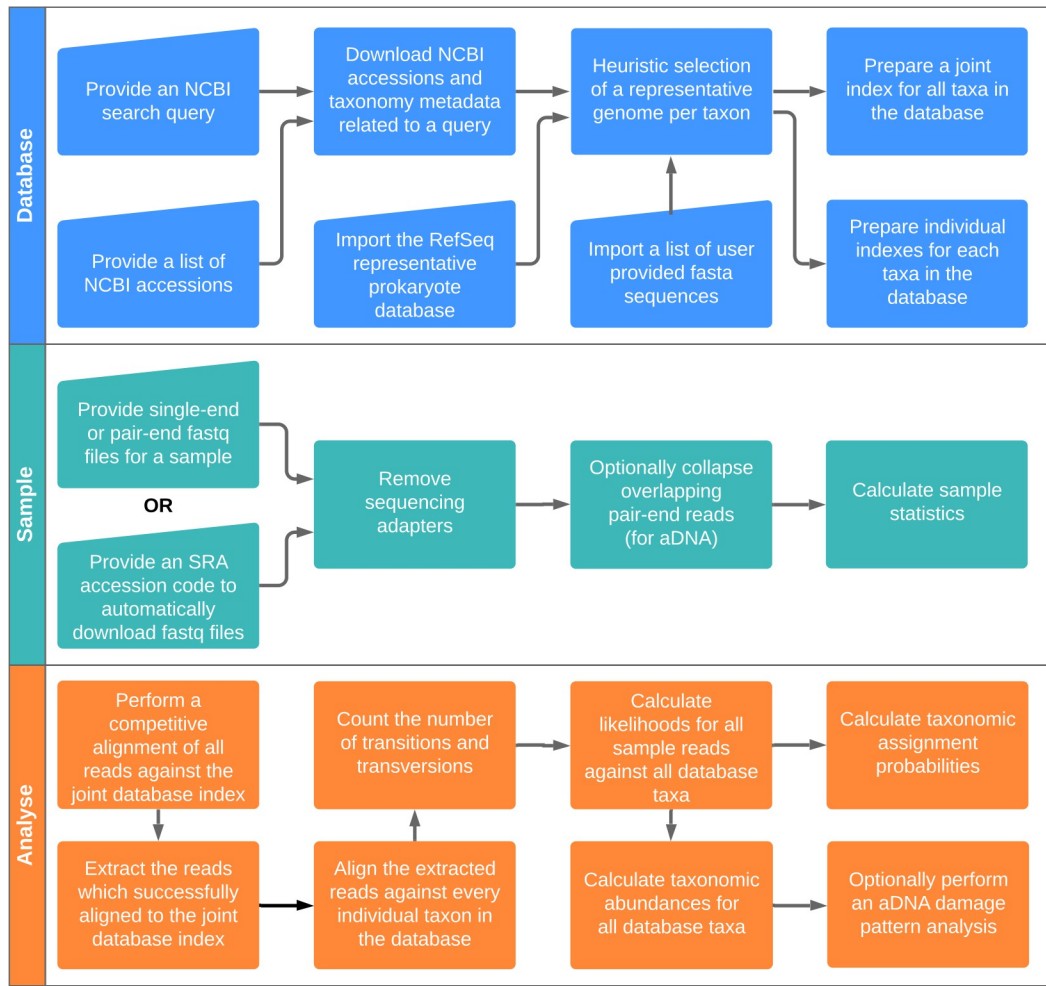

**Fig 1. HAYSTAC's workflow.** HAYSTAC consists of three main modules: (i) DATABASE, which builds a database of reference genomes from various user input sources; (ii) SAMPLE, which handles downloading sequencing files from the SRA and pre-processing of samples prior to analysis; and (iii) ANALYSE, which performs an analysis of a sample against a database by applying the mathematical model (see methods) for taxonomic abundance estimation.

A database-wide index is then built, followed by a competitive read mapping (very fast local alignment mode in Bowtie2, with default parameters) of each sample against the full database. This step ensures that reads with no matches in the database are excluded in all subsequent analyses ("Unknown Source" category).

Reads passing this initial inclusion filter are then aligned against every individual genome in the database (with Bowtie2 in end-to-end mode). For each alignment, we count the number of transitions ($T_s$) and transversions ($T_v$), as each type of mismatch contributes differently to our statistical model, in order to handle ancient DNA damage. We then compute a posterior probability for each read/taxon alignment pair, using both mismatch count and expected mismatch probability ($\sigma$, default of 0.05). Accuracy can be improved by adjusting the mismatch probability ($\sigma$) using prior information on intra- and interspecific variability in the specific set of species analysed.

Reads that possess a posterior probability exceeding a user-defined threshold (default of 0.75, S5 and S6 Figs) are then assigned to a single taxon. Reads with a lower posterior probability are assigned to the "ambiguous source" category. For each taxon, the count of reads passing the posterior probability threshold are used to compute the mean posterior abundance (and 95% confidence interval); *i.e.*, the relative abundance of a taxon in a sample, using a Dirichlet distribution of the sample input data as a whole. A final validation step, for low depth sequencing data, using breadth of coverage is then used to assess if the assigned reads represent a random genome sample or cluster around specific genomic regions.

## Computational performance

We assessed the computational performance of HAYSTAC relative to Sigma, Kraken2/Bracken, KrakenUniq and MALT by measuring peak memory usage (maximum resident set size) and elapsed execution time (wall clock) for each software package on a machine with 48 CPU cores and 392 GB of RAM under controlled conditions. To control for variability in each measurement, we ran five independent replicates of each job, and performed a linear regression across the observed values.

We first assessed performance when building databases containing 10, 100 and 500 species. For all tested sizes, HAYSTAC uses less memory than MALT, Kracken2/Bracken or KrakenUniq, as well as being faster than KrakenUniq for all database sizes and faster than MALT and Kracken2/Bracken for the 10 and 100 species databases and only marginally slower than MALT and Kracken2/Bracken for the 500 species database. For all databases HAYSTAC was slower and used more memory than Sigma, which does not require large indices for its alignments (Fig 2; S9 Table).

We then assessed the performance of these methods when analysing a one million read dataset against the same three database sizes. HAYSTAC uses less memory than Kraken2/Bracken, KrakenUniq and MALT for all database sizes, but it uses more memory and is slower than SIGMA for all database sizes (Fig 2; S10 Table).

Lastly, we assessed the performance of these methods when analysing datasets of 10 thousand, 100 thousand and 1 million reads, against a database of 500 species (S1 Fig; S10 Table). Memory performance was unchanged from the earlier benchmarks, showing that for all methods, peak memory usage scales with database size rather than sample size. With the exception of Kraken2/Bracken and KrakenUniq, all other methods saw only a modest increase in runtime with larger input sizes, showing that database size also has a substantial impact on overall runtime.

## Analyses of simulated data sets using RefSeq prokaryotic database

To characterise the sensitivity and specificity of HAYSTAC, we compared its performance to that of three other widely used metagenomic profiling tools: Sigma, Kraken2/Bracken,

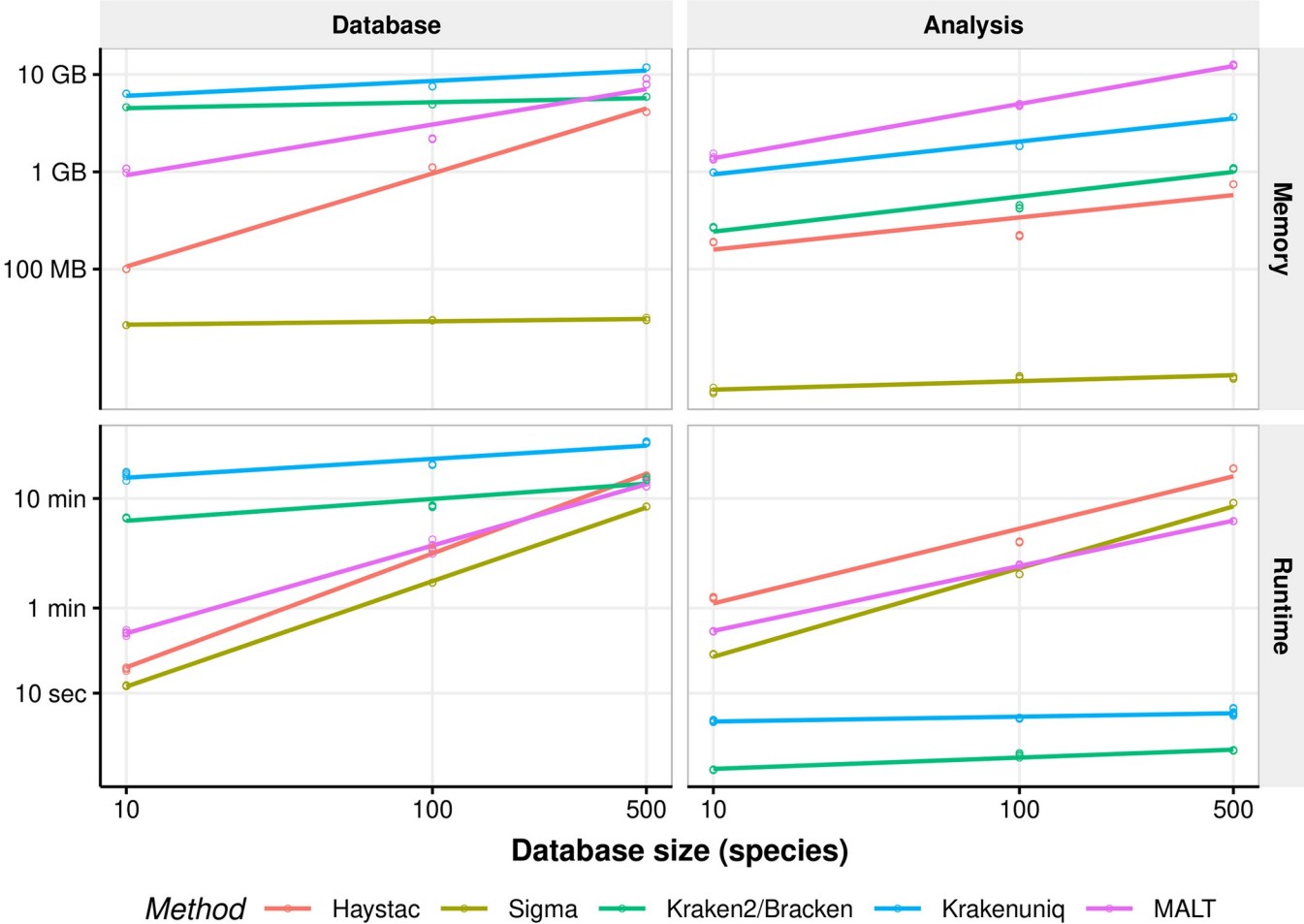

**Fig 2. Computational performance of HAYSTAC and other methods.** Linear regression of the elapsed time (wall clock) and peak memory usage (maximum resident set size) for a sample of size 1 million reads and reference databases containing either 10, 100 or 500 genomes, each with 5 replicates. When constructing the database, HAYSTAC uses substantially less memory and runs faster than either Kraken2/Bracken, KrakenUniq or MALT for restricted database sizes. When performing analyses, HAYSTAC uses less memory than Kraken2/Bracken, KrakenUniq or MALT, while its runtime was only marginally slower.

KrakenUniq and MALT. We simulated four datasets for these comparisons (Table 1), and then profiled each dataset with all five tools. Each simulated dataset was designed to test a different aspect of performance. For each test, we computed: (i) false positive count (total number

**Table 1. Characteristics of the six simulated samples sets of the Simple, Random and Oral microbiome datasets.**

| Simulation set | Dataset | Number of replicates | aDNA damage | Mean number of species |
|---|---|---|---|---|
| Simple with human DNA | Simple Microbiome | 2 | No | 10 |
| Simple without human DNA | Simple Microbiome | 2 | No | 10 |
| Microbiome 100 species ancient | General Microbiome | 2 | Yes | 100 |
| Microbiome 100 species modern | General Microbiome | 2 | No | 100 |
| Microbiome 500 species ancient | General Microbiome | 2 | Yes | 500 |
| Microbiome 500 species modern | General Microbiome | 2 | No | 500 |
| Oral microbiome aDNA damage | Oral Microbiome | 6 | Yes | 178 |
| Oral microbiome modern | Oral Microbiome | 6 | No | 178 |

of species that were reported yet not present in the simulated sample); (ii) false negative count (total number of species present in the simulation yet were not identified); and (iii) true positive count (total number of species correctly identified).

We first generated a dataset of samples comprising 10 species to quantify the ability of each program to accurately detect species in a simple metagenomic community (hereafter "Simple Microbiome" dataset). We generated two additional sets of profiles for this dataset: one that included modern human DNA ("contaminated") and one that did not ("noncontaminated"). This dataset was created using gargammel [36] by randomly sampling 10 species from a modified RefSeq prokaryote representative genome database (see Methods) and building simulated libraries of one million single-end reads with an equal number of reads per species (100K), both with and without contamination from the human reference genome (25% of the total simulated reads).

All tools, apart from KrakenUniq, correctly identified the 10 species in each simulated sample set (both with and without human contamination; Fig 3). KrakenUniq in both sample sets identified 9.5 species. Kraken2/Bracken (average of 2.5 species), KrakenUniq (average of 0.5 species) and MALT (average of 0.5 species) programs reported false positives in the non-contaminated sample set. False positive species counts, however, dramatically increased in the sample set with human contamination. HAYSTAC reported the lowest false positive count (5) relative to Sigma (6), Kraken2/Bracken (12.5), KrakenUniq (10.5) and MALT (12.5) (Fig 3). This suggests that HAYSTAC is less prone to false positive identifications caused by human contamination in bacterial reference genomes found in the RefSeq representative prokaryotic database. All false positive identification (across all programs) involved species whose RefSeq

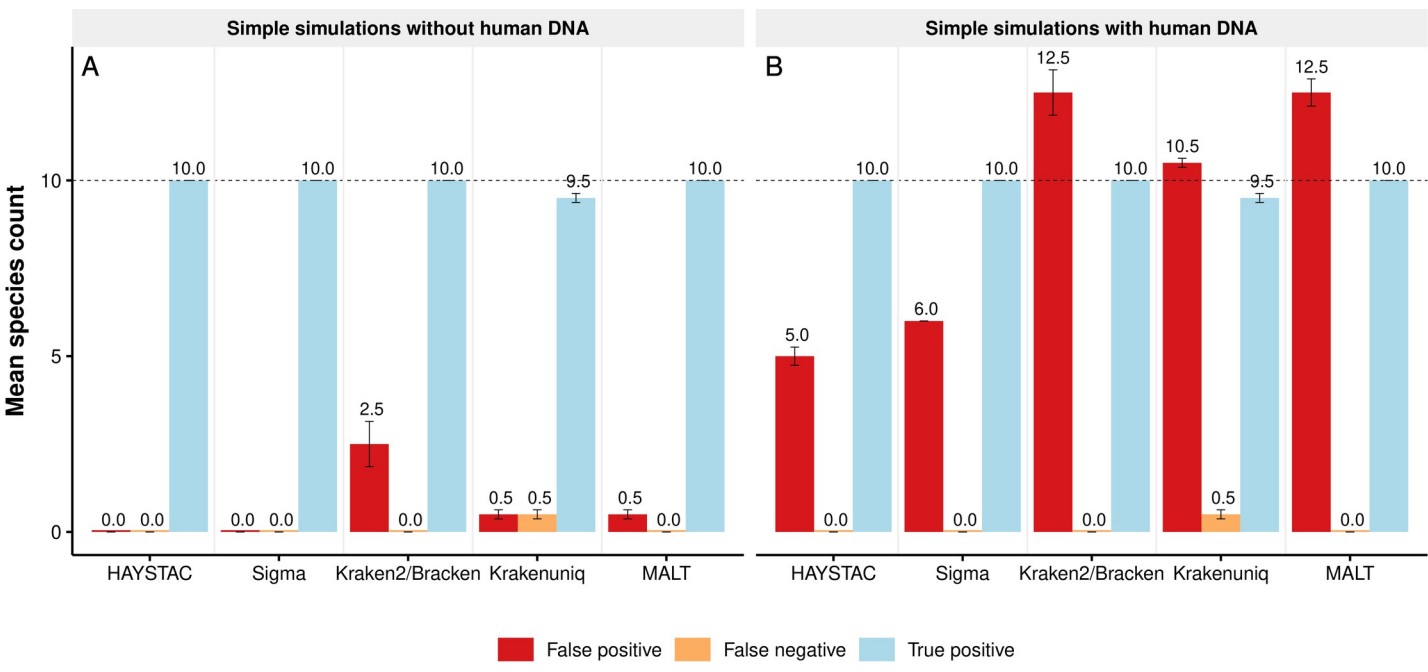

**Fig 3. Accuracy of HAYSTAC and other methods for a simple simulation.** Bar plot showing the mean count of false positives (red), false negatives (orange), and true detected species (blue) for two versions of the simple simulations dataset, each with two replicates: (A) without human DNA contamination (*n* = 2); and (B) with human DNA contamination (*n* = 2). The dotted line shows the number of simulated species in each set of samples (*i.e.*, the maximum true positive; *n* = 10), and numbers above the error bars indicate the mean species count in each category. For the simulation without human contamination, HAYSTAC outperforms Kraken2/Bracken, KrakenUniq and MALT, and performs equally with Sigma (*i.e.*, no false positives or false negatives). For the simulation with human contamination, HAYSTAC outperforms all four other methods. All of the 5 false positive species identified by HAYSTAC are known to contain human sequences in their reference genomes [37], confounding any analyses which do not explicitly filter human contamination.

genomes are known to contain human DNA contamination [37] (S7 Table). This result strongly supports the pre-filtering and removal of all human reads prior to microbial metage-nomic profiling.

We then tested HAYSTAC's performance when profiling samples with damage patterns characteristic of aDNA (*i.e.*, sequence fragmentation and cytosine deamination). To do so, we generated eight simulated samples consisting of 10 million reads each, both with and without aDNA damage (hereafter "General Microbiome" dataset). Each sample contained simulated reads from either 100 species (four simulated samples; two random sets of species all with equal abundance) or 500 species (four samples; two random sets of species all with equal abundance) that were randomly sampled from the RefSeq representative prokaryotic database (Table 1).

Overall, the incidence of false negatives was low. HAYSTAC reported 2.0 false negative species on average in the 100 species sample set, and 14.0 in the 500 species sample set. Sigma reported 2.0 and 15.5, Kraken2/Bracken 2.75 and 23.5, KrakenUniq 12.5 and 71, and MALT 8.0 and 40.75, respectively (S2 and S3A & S3B Figs). False positives, in contrast, were variable between methods. HAYSTAC reported an average of only 0.5 false positives in the 100 species sample set, and an average of 1.0 false positives in the 500 species sample set. Sigma performed similarly, with an average of 0.5 and 1.0 false positives, while Kraken2/Bracken reported 11.5 and 17.0 false positives, KrakenUniq reported 21 and 64.3 false positives and MALT reported 8.0 and 8.75 false positives in the 100 and 500 species sample sets, respectively (S3A and S3B Fig and S3 Table).

We next performed more realistic simulations based on species compositions that mimic an oral microbial community from a previously analysed subgingival plaque sample ([30]; hereafter "Oral Microbiome" dataset). These simulations included 12 simulated samples of 5 million paired-end reads each, with and without aDNA damage, and with 176–180 species in varying or constant species abundance (Table 1). Out of the 196 species that were simulated for the Oral Microbiome dataset, only 115 were included in the RefSeq representative prokary-otic database. From these 115 species, only 4 species were simulated from the same genomes that were present in the modified representative prokaryotic RefSeq database we used (S6 Table).

As expected, given the incompleteness of the database, false negatives were elevated overall (Table 2). HAYSTAC failed to identify on average 73.2 species in the ancient (damaged) Oral Microbiome dataset, and 73.2 species in the modern Oral Microbiome dataset. Sigma failed to identify on average 74.2 and 74.2 species, Kraken2/Bracken failed to identify 74.0 and 73.8 spe-cies, KrakenUniq failed to identify 79.2 and 78.0 species and MALT failed to identify 75.0 and 74.8 species in the ancient and modern Oral Microbiome datasets, respectively (Figs 4A and 4B and S4). False positives, however, were highly variable between methods. HAYSTAC

**Table 2. False negative and false positive species counts in the ancient and modern Oral Microbiome datasets.**

|  | Oral microbiome | | Oral microbiome | |
|  | aDNA damage | | modern | |
|  | False | False | False | False |
|  | negative | positive | negative | positive |
|---|---|---|---|---|
| HAYSTAC | 73.2 | 29.7 | 73.2 | 35.3 |
| Sigma | 74.2 | 42.2 | 74.2 | 29.5 |
| Kraken2/Bracken | 74.0 | 170.7 | 73.8 | 224.2 |
| KrakenUniq | 79.2 | 62.0 | 78.0 | 134.2 |
| MALT | 75.0 | 137.2 | 74.8 | 157.5 |

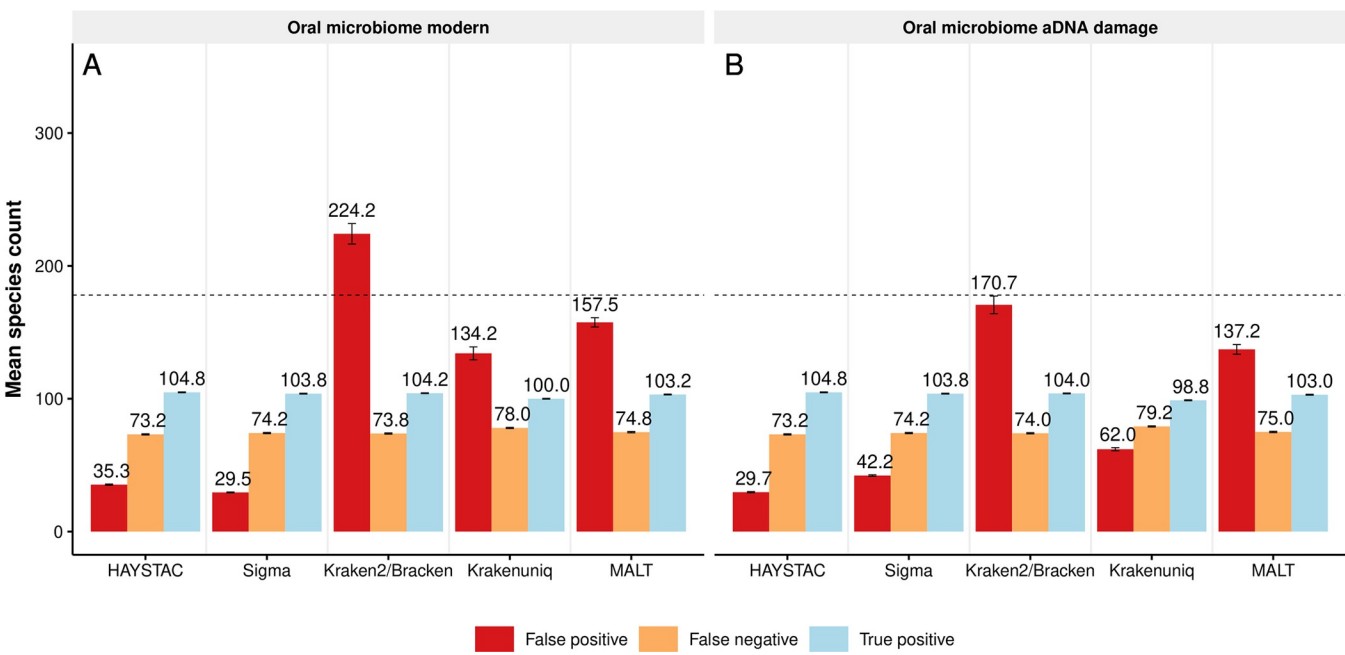

**Fig 4. Accuracy of HAYSTAC and other methods for an oral microbiome simulation.** Bar plot showing the mean count of false positives (red), false negatives (orange), and true detected species (blue) for two versions of the oral microbiome dataset, each with six replicates: (A) modern simulation, with fixed read lengths ($n = 6$); and (B) ancient simulation, with variable read lengths and post-mortem damage ($n = 6$). The dotted line shows the average number of simulated species in each set of samples (*i.e.*, the maximum true positive; $n = 178$), and numbers above the error bars indicate the mean species count in each category. For the modern simulation, HAYSTAC substantially outperforms Kraken2/Bracken, KrakenUniq and MALT with respect to false positives, and performs equivalently with Sigma. For the ancient simulation, HAYSTAC outperforms all four other methods with respect to false positives. The overall high rates of false negative identifications are due to the absence of many simulated species from the reference database for all four methods. HAYSTAC also outperforms all the other four methods in both the modern and ancient Oral Microbiome datasets by identifying the highest number of true positive species.

reported an average of 29.7 false positives in the ancient Oral Microbiome dataset and 35.3 false positives in the modern Oral Microbiome dataset. Sigma reported 42.2 and 29.5 false positives, Kraken2/Bracken reported 170.7 and 224.2 false positives, KrakenUniq reported 62.0 and 134.2 false positives and MALT reported 137.2 and 157.5 false positives in the ancient and modern datasets, respectively (Fig 4A and 4B and S4 Table). Overall HAYSTAC outperformed the other methods in the ancient Oral Microbiome sample set with the lowest number of false positive identifications, and the highest number of true positive identifications in both the ancient and modern Oral Microbiome datasets.

## Analyses of simulated data sets using genus-level database

We next assessed how accurately HAYSTAC identified specific candidate species using smaller reference databases. To do so, we re-analysed 20 simulated samples included in the General and Oral Microbiome datasets (General Microbiome 100 species, General Microbiome 500 species, Oral Microbiome aDNA damage, and Oral Microbiome modern datasets) (Table 1) using nine different genus-specific databases (see Methods). For each randomly selected genus, we constructed a database that included all genomes found in the RefSeq representative prokaryotic database. The nine genera that were selected for this analysis were: *Bacteroides*, *Burkholderia*, *Campylobacter*, *Clostridium*, *Corynebacterium*, *Desulfitobacterium*, *Mycobacterium*, *Solimonas*, and *Streptococcus*. Each database consisted of all RefSeq genomes from all species belonging to the genus instead of the full RefSeq representative prokaryotic database. This resulted in databases with between 3–91 species each, as opposed to 5,652 species when

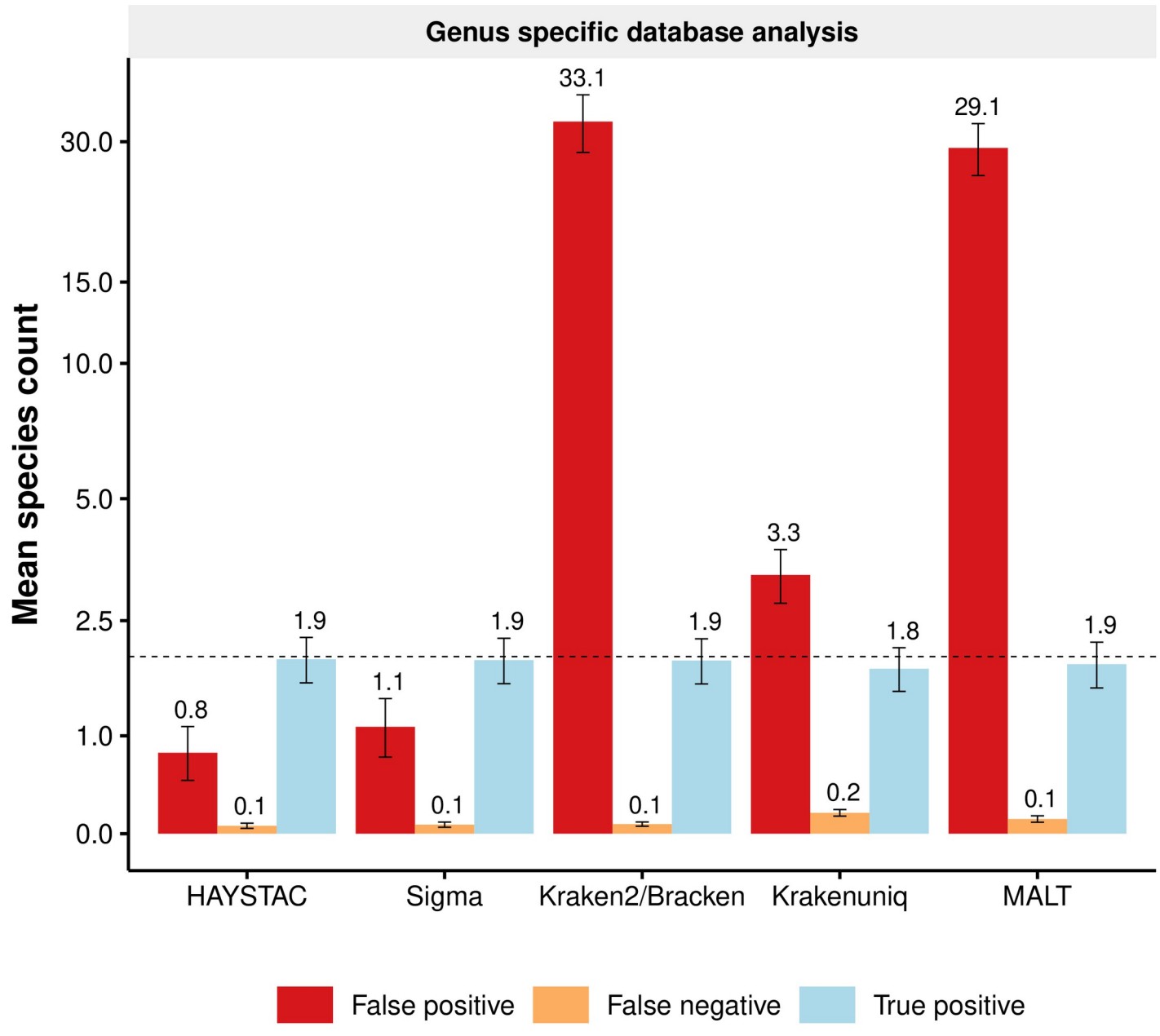

**Fig 5. Accuracy of HAYSTAC and other methods using a reference database restricted to a single genus.** Bar plot, with a pseudo-log10 transformed y-axis, showing the mean count of false positives (red), false negatives (orange), and true detected species (blue) for nine different genera (*Bacteroides*, *Burkholderia*, *Campylobacter*, *Clostridium*, *Corynebacterium*, *Desulfitobacterium*, *Mycobacterium*, *Solimonas* and *Streptococcus*), each with 20 samples from the general and oral microbiome datasets. The dotted line shows the average number of simulated species in each set of samples (*i.e.*, the maximum true positive; *n* = 2.0), and numbers above the error bars indicate the mean species count in each category. For the genus specific analysis, HAYSTAC substantially outperforms both Kraken2/Bracken and MALT with respect to false positives and performs better than Sigma.

using the full RefSeq representative prokaryotic database. For each analysis, we computed false positive, false negative, and true positive species counts as before.

Overall, false negatives were low. HAYSTAC reported on average 0.1 species missing in the samples of the General and Oral Microbiome datasets, while Sigma reported 0.1, Kraken2/Bracken 0.1, KrakenUniq 0.2 and MALT 0.1 species missing (Fig 5, S5 Table). False positives,

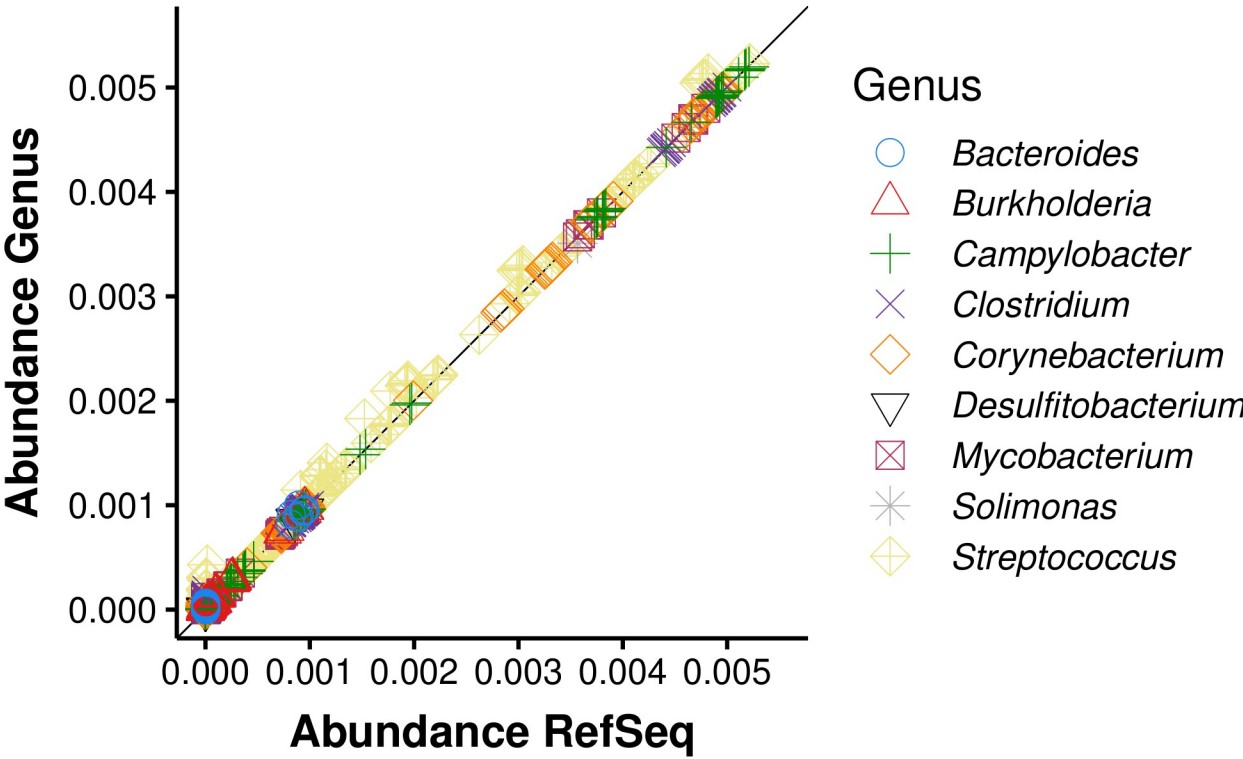

**Fig 6. HAYSTAC inferred posterior abundance levels.** Scatter plot showing the mean posterior abundances across all taxa ($n$ = 362) and samples ($n$ = 20) for either a genus specific database or the entire RefSeq representative database of prokaryotic species. Using a genus specific database has a small positive bias in mean posterior abundance for taxa within that genus (paired t-test p-value < $2.2 \cdot 10^{-16}$, mean of the differences = $3.9 \cdot 10^{-6}$), nevertheless the overall abundance levels are highly correlated ($R^2$ = 0.999). Computational runtime for the genus specific analyses are faster and use less memory, making genus specific analyses suitable for rapid initial screening (e.g. 1 million reads against a *Corynebacterium* specific database runs approximately 6.15 faster than against a database containing 500 species and uses approximately 3.9 times less memory).

however, were variable between methods. HAYSTAC reported an average of 0.8 false positives, averaged across all samples in the General and Oral Microbiome datasets, while Sigma reported 1.1, Kraken2/Bracken 33.1, KrakenUniq 3.3 and MALT 29.1 (Fig 5). We also used HAYSTAC to compare the mean posterior abundance computed using the full RefSeq database with abundance computed using the genus specific databases (Fig 6). We found that while genus specific abundances were in cases slightly higher than those computed with the full RefSeq database, there was a very strong correlation between the two ($R^2$ = 0.999; slope = 1; p-value = 0; GLM: $y = x + 3.1 \cdot 10^{-6}$).

## Pathogen identification from metagenomic communities

We then tested the ability of HAYSTAC and HOPS (an extension of MALT) to distinguish specific human pathogen species from close relatives when occurring at low abundance in a metagenomic sample. To do so we selected 100 (human) pathobionts, each one belonging to a different genus, as well as 1 non-pathogenic microbial taxon from each of the selected genera (total of 200 taxa, 2 per genus, S11 Table). We subsequently simulated a total of 200 libraries (1 library per species), each containing 100 reads with damage patterns characteristic to aDNA. This data was then added to one of our previously simulated Oral Microbiome dataset samples. This resulted in 200 replicates of an Oral Microbiome sample (anc200e2repgn), with each replicate containing 1 human pathobiont at low abundance. Due to the lack of taxonomic specificity present in a random sample of only 100 reads per pathogen genome, the overall true

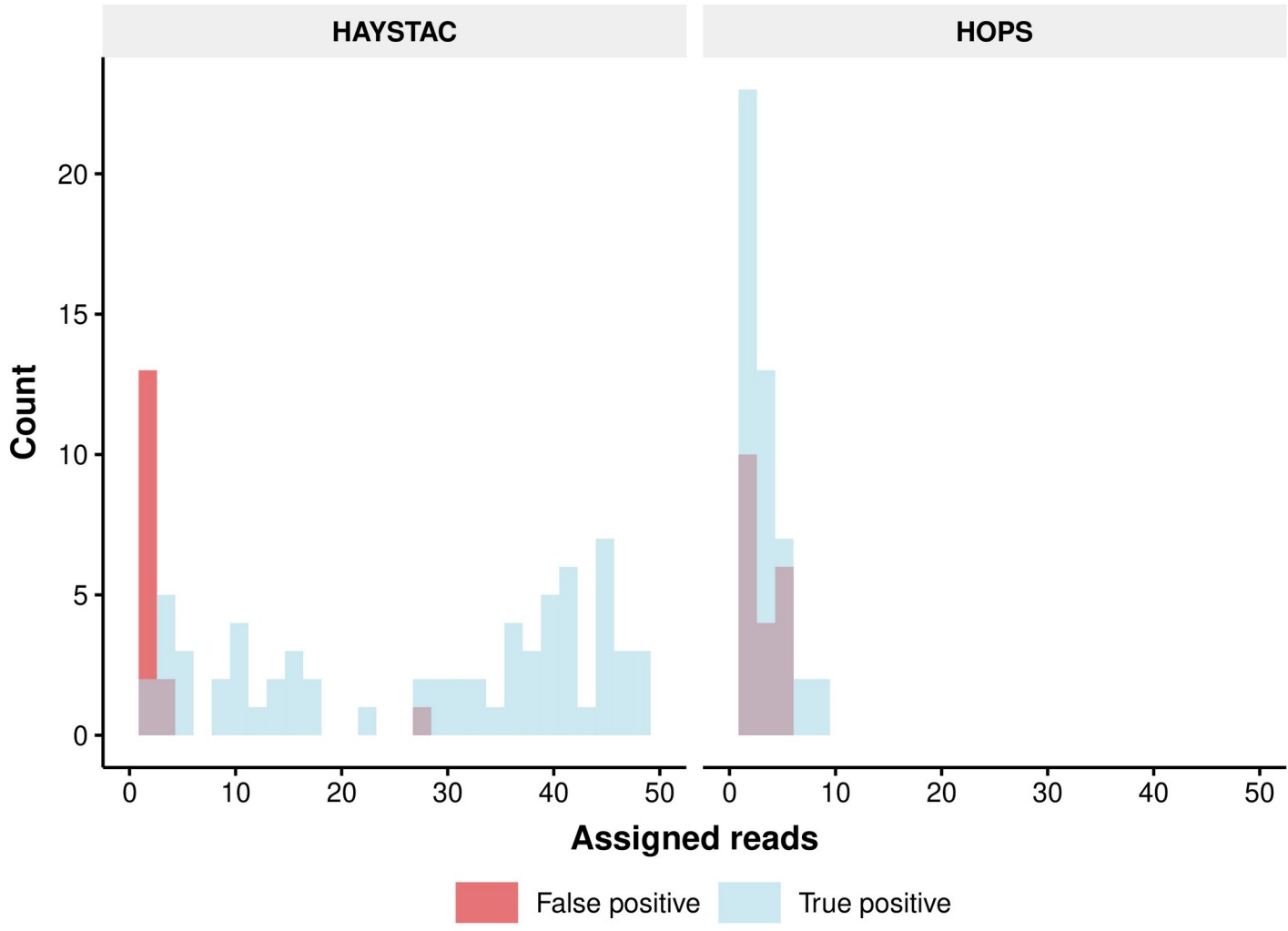

**Fig 7. Histogram of the number of assigned pathobiont simulated reads.** Histogram showing the read count frequency of true positive (blue) and false positive (red) pathobiont reads as identified by HAYSTAC and HOPS, after screening 200 spiked iterations of an ancient Oral Microbiome dataset sample (anc200e2repgn). HAYSTAC identifies robustly more pathobiont reads than HOPS, while producing less false positive identifications.

positive rate for this test was low. HAYSTAC identified 2016 true positive pathogenic reads and misidentified 50 reads across the 200 replicates, while HOPS only identified 144 true positive pathogen reads, and misidentified 58 reads across all replicates (Fig 7).

## Case study 1: Ancient *Yersinia pestis*

We then tested HAYSTAC's performance to detect specific pathogens in real archaeological samples. We first analysed 130 publicly available plague-positive ancient human libraries [9], obtained from seven bone and tooth samples (RISE00, RISE139, RISE386, RISE397, RISE505, RISE509 and RISE511).

HAYSTAC was run using a reference database consisting of the longest complete genomes for each of the species in the genus *Yersinia* (including plasmid sequences). Our method was able to confidently assign between 6,720 and 856,467 reads to *Y. pestis* across all seven samples (Fig 8). HAYSTAC on average uniquely assigned an order of magnitude fewer reads compared to the number of aligned reads reported in the original study [9]. In addition, we found evidence for additional *Yersinia* species in all samples, including *Y. pseudotuberculosis* (606–

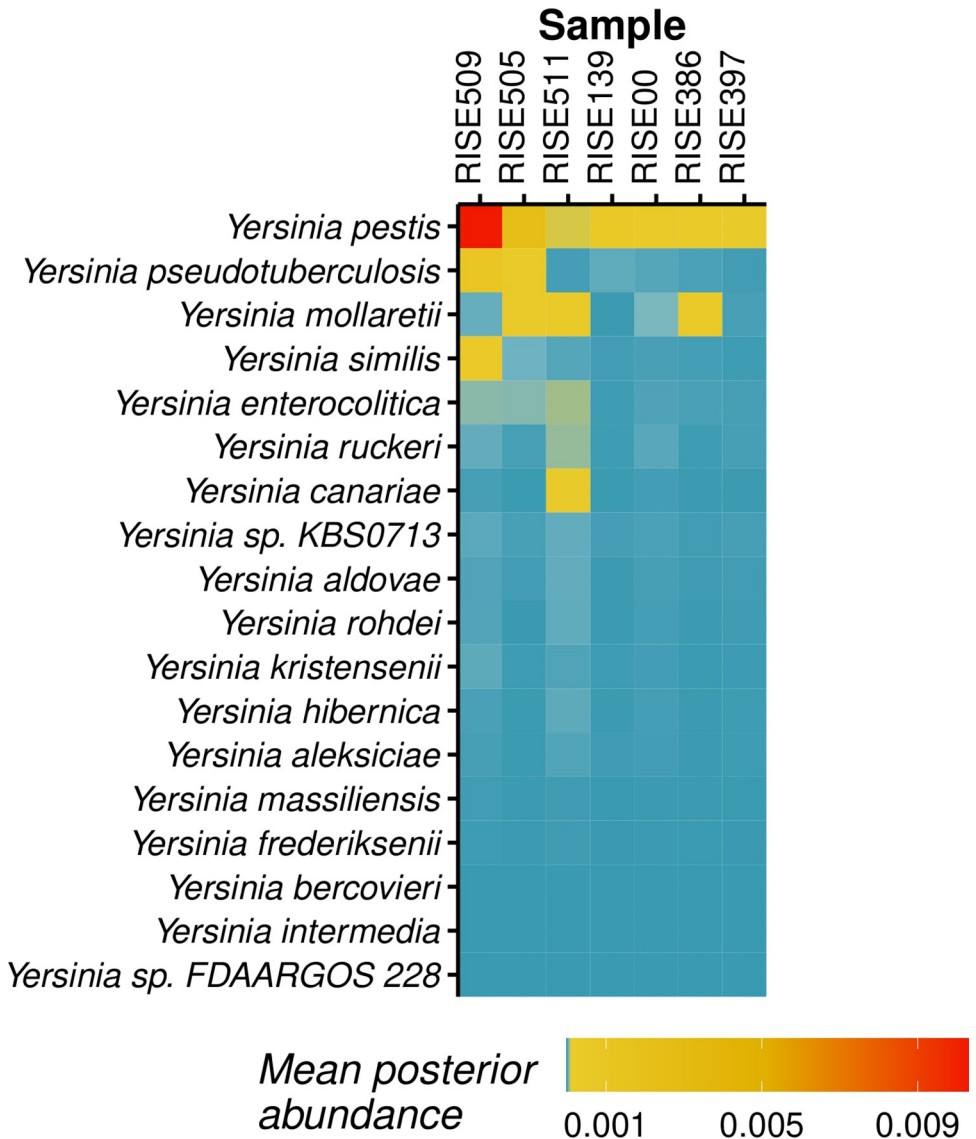

**Fig 8. Posterior abundances of *Yersinia* species in Case Study 1.** Heatmap showing the mean posterior abundances for the seven RISE samples, based on a genus specific analysis of 18 *Yersinia* species. *Yersinia pestis* is the species with the highest posterior abundance, followed by *Y. pseudotuberculosis*, in agreement with the results of (Rasmussen et al., 2015).

72,581 reads), *Y. similis* (376–26,010 reads), *Y. canariae* (74–8,868 reads), *Y. enterocolitica* (209–5,040 reads), *Y. mollaretii* (147–24,241 reads) and *Y. ruckeri* (211–4,670 reads). Of these, *Y. ruckeri* and *Y. similis* are known to cause infections in animals (specifically fish), but have also been reported to infect humans [38]. These identifications, however, are likely spurious, since *Y. ruckeri* and *Y. similis* are phylogenetically basal to other *Yersinia* species [39,40] and may be attracting reads belonging diverged sequences.

The posterior abundances for *Yersinia pestis* in the RISE samples ranged from 0.0093–1.02%, corresponding to between 6,720–856,467 assigned reads. Despite the large number of assigned reads, posterior abundances only exceeded 0.01% for six of the seven samples (RISE00, RISE139, RISE386, RISE397, RISE505 and RISE509). This highlights the need to consider both relative and absolute abundance when making positive identifications.

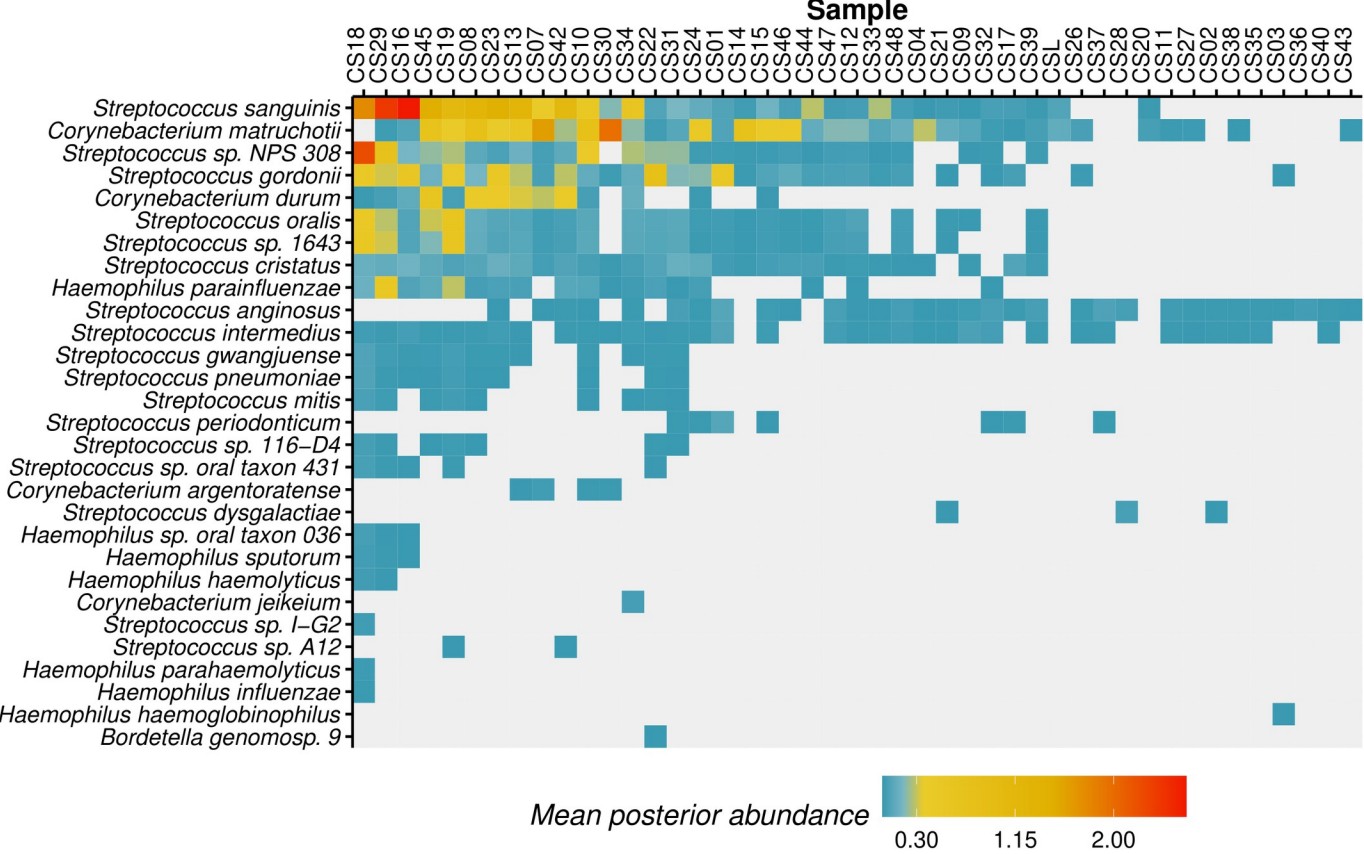

**Fig 9. Posterior abundances of oral microbiome species in Case Study 2.** Heatmap showing the mean posterior abundances for the 44 dental calculus samples, based on a custom database that combined the prokaryotic representative RefSeq and pathobionts with complete genomes from the following 5 genera: *Corynebacterium*, *Haemophilus*, *Klebsiella*, *Streptococcus*, and *Bordetella*. Species from these five genera of interest that naturally colonise the oral cavity can be found in more samples and at higher abundance (e.g. *C. matruchotii*) compared to pathobionts of the upper respiratory system (e.g. *S. pneumoniae*).

## Case study 2: Historic dental calculus

Dental plaque is a biofilm that develops naturally on teeth, and during life it periodically mineralizes to form dental calculus [41], a robust substrate that can preserve biomolecules such as DNA from the biofilm and the host [34]. Although this substrate can act as a reservoir for respiratory pathobionts, analysis of ancient dental calculus metagenomic data is often challenging because the low abundance of these species makes it difficult to confidently identify them [34]. To test the performance of HAYSTAC on pathobiont aDNA sequences within host-associated ancient microbiomes, we analysed data from 48 dental calculus samples from a 19[th] century hospital in Oxford [42,43].

We screened these libraries for potential upper respiratory pathobionts from the *Corynebacterium*, *Streptococcus*, *Klebsiella* and *Bordetella* genera (Fig 9), in an effort to test if such pathobionts could be preserved and identified in dental calculus. Members of the genera *Corynebacterium*, *Streptococcus* and *Bordetella* had also been included in the Oral Microbiome dataset. Within the *Corynebacterium* genus, we identified *Corynebacterium matruchotii* (abundance 0.01–1.96%; 37 samples) and *C. durum* (abundance 0.01–0.59%; 15 samples) both of which are oral commensal taxa. We did not, however, identify *Corynebacterium diphtheriae*, the species that causes diphtheria. We identified multiple species of *Streptococcus*, with the commensal *Streptococcus sanguinis* being identified in 33 samples at relatively high abundance

(0.01–2.62%). The pathobiont *Streptococcus pneumoniae* was also positively identified at low abundance (0.01–0.07%) in 10 samples. We did not find any evidence for species belonging to the genera *Klebsiella* and *Bordetella*.

Members of the *Haemophilus* genus were rarely identified, which is expected given that many species from that genus are generally low-abundance in dental plaque compared to other oral surfaces such as the tongue [44], and are similarly low-abundance in dental calculus [43]. However, *Haemophilus parainfluenzae* was found in 18 samples (abundance between 0.01–0.46%), *H. haemolyticus* was identified in two samples (abundance 0.01–0.03%), and *H. parahaemolyticus* was identified only in sample CS18 (abundance 0.02%). An uncharacterized oral species, *Haemophilus oral taxon 036*, was identified in three samples (abundance 0.01–0.04%) and the commensal *H. sputorum* was also identified in three samples (abundance 0.01–0.03%).

As expected, our analysis indicates that while species naturally occurring in the oral cavity (e.g., *C. matruchotii* and *durum)*, were detected in most samples, upper respiratory system pathobionts (e.g. *S. pneumoniae*, *H. parainfluenzae*) were more rare. Although pathogens were found generally at very low abundance, in only a few individuals, our results indicate that HAYSTAC was able to positively identify some potentially pathogenic species such as *Streptococcus pneumoniae* and *Haemophilus parainfluenzae*, suggesting that dental calculus is a promising substrate to study ancient respiratory pathobionts.

## Discussion

Our results demonstrate that HAYSTAC provides a robust and computationally efficient framework to statistically assess the presence of specific species in a metagenomic sample. HAYSTAC identified fewer false positives than Kraken2/Bracken, KrakenUniq or MALT in all simulations, and fewer than Sigma in simulations of ancient data. When analysing modern data, HAYSTAC's performance can be further optimised by reducing the base mismatch probability parameter ($\sigma$) at run-time leading to a more stringent alignment strategy, and increasing the minimum relative abundance threshold for a species call. Especially reducing the base mismatch probability ($\sigma$) will reduce the number of false positive identifications, because HAYSTAC calculates the maximum number of mismatches for a valid alignment based on both the base mismatch probability ($\sigma = 0.05$) and the average read length of reads in a library (e.g., 150 bp for the Oral Microbiome modern dataset, which allows for more mismatches to be allowed). It uses less memory than Kraken2/Bracken, KrakenUniq and MALT. Furthermore, our results show that the use of reduced reference databases (e.g., a few selected genera as opposed to the entire RefSeq representative prokaryotic database), can dramatically reduce computational time (approximately 6.15 times faster when using a database of 84 species than a database of 500) while still providing highly accurate species identification. HAYSTAC's underlying computational framework allows for an easy installation of the package, and the efficient maintenance of its source code and software dependencies. Altogether, this makes HAYSTAC a flexible tool to rapidly and accurately profile metagenomic samples and screen them for pathogens, a feature that is attractive for a large range of applications, including archaeogenetics and biosurveillance.

### Species concept, horizontal gene transfer and species identification

Although we demonstrated that both HAYSTAC and Sigma provide more robust species assignments than MALT, Kraken2/Bracken or KrakenUniq, their internal computation is based on the assumption that intraspecific polymorphism is lower than interspecific polymorphism. While this is true in most cases, the concept of a species among microorganisms is

often not straightforward. This is the case, for example, in the *Mycobacterium tuberculosis* complex in which interspecific variability is very low [45], and occasionally lower than the default value of expected mismatch (*i.e.*, expected level of intraspecific polymorphism) set in HAYSTAC ($\sigma$ = 0.05; see Eq 4 below). HAYSTAC deals with this issue by introducing an "ambiguous source" category, to which reads that are equally likely to match two or more genomes are assigned. This ensures that they do not contribute to posterior abundance computations, which could possibly force false identifications, while also not dismissing them into "unknown source" category with reads that cannot be classified. In cases where the levels of intra- and interspecific diversity are known (e.g., within a genus) it is possible to adjust the probability of the expected mismatches between a read and a genome ($\sigma$; see Eq 4), which can increase power while reducing false positive rates when analysing a specific dataset. This feature means HAYSTAC's performance can be finetuned to specific questions and datasets.

Identifying species within a metagenomic sample can also be complicated by horizontal gene transfer (HGT). HGT can lead to spurious, highly supported, species assignments—especially if the sequences originate from an outgroup species that is not represented in the database. HAYSTAC mitigates this issue by inspecting the evenness of coverage across the genome for shallow sequencing data. Specifically, reads that belong to an outgroup species and that map to a genome in a reference database because of an HGT or due to contamination are expected to map only to parts of the reference genome (e.g. the part that was horizontally transferred). HAYSTAC assesses whether reads are clustered more than expected by chance across a reference genome using an evenness of coverage metric (see Methods). This provides the user with a quantification of the evenness of spread, and thus a proxy signature of potential HGT or misassignment of reads from a taxon that is not represented in the reference database.

## Issues with databases

Another issue facing metagenomics is contaminated sequences incorporated into reference genomes. For example, multiple bacterial RefSeq representative assemblies contain human sequences [37]. This can lead to spurious, highly supported, species assignments when profiling metagenome samples. Including genome sequences from potential contaminant sources (e.g., the human reference genome) in the database should alleviate these issues [46]. Alternatively, it is also possible to filter reads before starting a taxonomic identification/assignment analysis (*i.e.*, by excluding those that map to the contaminant's reference genome). Potential contaminants, however, are not always known, and there are many gaps in reference sequence databases. In fact, inadequate species representation in databases is likely to be an important source of false positives for all species identification methods [15].

Although reads belonging to a species that is not represented in a database are typically assigned to a higher taxonomic node ("ambiguous source" category in HAYSTAC), incorrect species assignment can occur if the missing taxon is closely related to another species in the database, or if the read derives from a conserved element found within a taxonomic group with sparse representation in the database. Although this is not considered a major problem for general profilers, as closely related species are likely to play a similar function in the ecosystem, such misassignments can greatly complicate the identification of specific pathogen species, especially if they have close environmental relatives.

## Conclusions

Metagenomic data are now routinely produced and analysed, providing new insights into microbial communities that were previously unknown. Development of metagenomic analytical tools, however, has mostly focused on frameworks that can profile an entire microbial

community. We showed that these methods, however, can produce high levels of false positives when focusing on specific species identification. Here we present a tool that can robustly assess whether a specific species is present in a metagenomic sample in one step, without the need to combine different pipelines to validate the results. HAYSTAC provides a user-friendly and automated method for rapidly constructing databases, and produces robust identifications regardless of the database size, allowing for both rapid hypothesis-driven analyses and exhaustive profiling. Because of the mathematical framework it employs, HAYSTAC reliably produces the lowest number of false positive identifications, making it a valuable tool for both ancient and modern DNA microbial research.

## Methods

**Single source identification.**    We collect a series of observations (reads:$r_1 \cdot r_n$) and we denote the full set of observations as $R$. We map all the reads to a database of reference genomes, and we define the event of sampling from the $i$-th reference genome in our database as $G_j$. Assuming that all reads come from the same source (*i.e.*, a single species represented in the database), the events of sampling from a given genome are exhaustive and mutually exclusive, *i.e.*

$$G_1 \cdots G_n | G_i \cap G_j = \emptyset, \quad \forall i \neq j \quad and \quad \bigcup_i^n G_i = \Omega$$

From Bayes Theorem we have:

$$P\Big(G_j|R\Big) = \frac{P(R|G_j)P(G_j)}{P(R)} = \frac{P(R|G_j)P(G_j)}{\sum_k P(R|G_k)P(G_k)} \tag{1}$$

Given that each read is a different realization of sampling process (*i.e.*, an independent observation), we can express the likelihood as:

$$P(R|G_j) = \prod_i P(r_i|G_j) \tag{2}$$

Therefore, the posterior probability of sampling from a given genome $G_j$ is:

$$P\Big(G_j|R\Big) = \frac{P(G_j)\prod_i P(r_i|G_j)}{\sum_k P(G_k)\prod_i P(r_i|G_k)} \tag{3}$$

## Likelihood

We next derive a likelihood function to express the probability of a read given a genome $P(r_i|G_j)$. To do so, we use a previously published expression of this likelihood [3] based on the number of mismatches between a read and a reference to which it is aligned:

$$P(r_i|G_j) = \sigma^z(1-\sigma)^{l-z}; \qquad z \leq U$$

$$P(r_i|G_j) = 0; \qquad z > U$$

where $z$ is the number of mismatches between read $r_i$ and genome $G_j$, $U$ is the maximum number of mismatches allowed in the alignment and $l$ is the length of read $r_i$ (in bp).

The method assumes a uniform probability of mismatch within a given alignment ($\sigma$) to account for population level variability and sequencing errors. We expand this framework to leverage the fact that transitions and transversions occur at significantly different rates, particularly in ancient samples due to the effect of deaminations.

For each alignment we can then express the likelihood of generating a read $r_i$ given that it has been sampled from the reference genome $G_j$ based on both number of transitions ($t$) and transversions ($v$).

Let $\sigma_t$ be the probability of observing a transition and $\sigma_v$ the probability of observing a transversion, we can express the likelihood for a single read as:

$$P(r_i|G_j) = \sigma_v^{v_{i|j}}\sigma_t^{t_{i|j}}(1 - \sigma_v - \sigma_t)^{l_i - v_{i|j} - t_{i|j}} \tag{4}$$

where $l_i$ represent the length of the read $r_i$, while $v_{i|j}$ and $t_{i|j}$ denote the number of transversions and transitions between the read $r_i$ and the genome $G_j$ respectively.

## Posterior

By substituting the likelihood term (Eq 4) into Eq 3, we can obtain an explicit form for the posterior probability of sampling a set of reads ($R$) from a given reference genome:

$$P\big(G_j|R\big) = \frac{\sigma_t^{\sum_i t_{i|j}}\sigma_v^{\sum_i v_{i|j}}(1 - \sigma_v - \sigma_t)^{\sum_i (l_i - t_{i|j} - v_{i|j})}P(G_j)}{\sum_{k=1}^{k=n} P(G_k)\sigma_t^{\sum_i t_{i|k}}\sigma_v^{\sum_i v_{i|k}}(1 - \sigma_v - \sigma_t)^{\sum_i (l_i - t_{i|k} - v_{i|k})}} \tag{5}$$

By rescaling both numerator and denominator by a factor of $(1 - \sigma_v - \sigma_t)^{\sum_i l_i}$ we get:

$$P\big(G_j|R\big) = \frac{P(G_j)\left(\frac{\sigma_t}{1-\sigma_v-\sigma_t}\right)^{\sum_i t_{i|j}}\left(\frac{\sigma_v}{1-\sigma_v-\sigma_t}\right)^{\sum_i v_{i|j}}}{\sum_{k=1}^{k=n} P(G_k)\left(\frac{\sigma_t}{1-\sigma_v-\sigma_t}\right)^{\sum_i t_{i|k}}\left(\frac{\sigma_v}{1-\sigma_v-\sigma_t}\right)^{\sum_i v_{i|k}}} \tag{6}$$

Let $V_j = \Sigma_i v_{i|j}$ denote the total number of observed transversions between all reads and genome $G_j$ while $T_j = \Sigma_i t_{i|j}$ represents the total number of transitions. Let $\delta_v = \frac{\sigma_v}{1-\sigma_v-\sigma_t}$ and $\delta_t = \frac{\sigma_t}{1-\sigma_v-\sigma_t}$ denote instead the transversion to match ratio and the transition to match ratio, respectively. By introducing this more compact notation, the previous equation can be re-written as:

$$P\big(G_j|R\big) = \frac{P(G_j)\delta_t^{T_j}\delta_v^{V_j}}{\sum_k^n P(G_k)\delta_t^{T_k}\delta_v^{V_k}} \tag{7}$$

## Numerical representation

Our method assumes a fixed probability of mismatches ($\sigma_v + \sigma_t$) which can be specified by the user and has a default value of 5%. We estimate the transition/transversion ratio based on the total number of mismatches per categories observed in the entire dataset and we obtain the value for $\sigma_v$ and $\sigma_t$ by solving the following linear system:

$$\begin{cases} \sigma_t + \sigma_v = 0.05 \\ \sigma_t = \dfrac{T}{V}\sigma_v \end{cases} \tag{8}$$

where $V = \sum_j\sum_i v_{i|j} = \sum_j V_j$ and $T = \sum_j\sum_i t_{i|j} = \sum_j T_j$.

If we assume a uniform prior in which each genome has a probability of $1/n$ we can simplify Eq 7 even further obtaining:

$$P\left(G_j|R\right) = \frac{\delta_t^{T_j}\delta_v^{V_j}}{\sum_k^n \delta_t^{T_k}\delta_v^{V_k}} \tag{9}$$

When analysing a large number of reads, evaluating Eq 9 might become problematic due to numerical representation issues. To accommodate for this we use an equivalent expression of the posterior probability:

$$P\left(G_j|R\right) = \frac{1}{1 + \sum_{k\neq j}^n \delta_t^{T_k-T_j}\delta_v^{V_k-V_j}} \tag{10}$$

Each term in the summation at the denominator of Eq 10 represents the likelihood of genome $G_k$ relative to genome $G_J$. There might be cases in which even the relative likelihood is so close to zero that cannot be represented numerically. We then take a conservative approach and assign to $\delta_t^{T_k-T_j}\delta_v^{V_k-V_j}$ the smallest number that the machine can represent (for a 64bit system it should be $2.2250738585072014\cdot 10^{-308}$ which we roughly approximate with $10^{-300}$).

## Multiple sources identification

We then expand this method to identify multiple species from metagenomic data. The multiple source identification approach involves the following steps:

- Assigning each read to a reference genome

- Obtain the likelihood for the multinomial distribution of abundances

- Obtain the posterior distribution from the Dirichlet distribution

- Calculate the 95% credible interval around each posterior mean

- Assess significance

Operatively we apply Eq 9 to each read separately and obtain a posterior probability for a read to originate from a given genome:

$$P\left(G_j|r_i\right) = \frac{\delta_t^{t_{i|j}}\delta_v^{v_{i|j}}}{\sum_k^n \delta_t^{t_{i|k}}\delta_v^{v_{i|k}}} \tag{11}$$

Thus, for each aligned read we obtain a vector of posterior probabilities (one value for each genome in our dataset). If a posterior probability value is above a given threshold (default: 0.75) we consider the read $r_i$ to be informative and assign it to that reference genome. If instead all posterior values are below the threshold, we consider the read to be uninformative and we assign it to the $n+1$ category where $n$ is the number of reference genomes. This means that any read $r_i$ that has been successfully aligned, can only be assigned to a single category: either to one reference genome or to the "Ambiguous Source" (AS, the $n+1$ category). All reads which belong to species that do not possess a representative reference genome (or a close relative) in the database and therefore haven't been successfully aligned to any of the reference genomes, will then be assigned to the $n+2$ category called "Unknown Source" (US).

Repeating this procedure $\forall r_i \in R$ allows us to populate a matrix $M_{N\times(n+2)}$ in which each row represents a read and each column represents a reference genome in our database or either the AS or US categories. Hence, the assignment matrix represents a series of $N$ observations from

a multinomial distribution of $n+2$ variables (where $N$ is the total number of reads). This distribution is parametrized by a probability mass function $q = [q_1, q_2, \ldots, q_n, q_{AS}, q_{US}]$.

The likelihood of $N$ observations has thus the following form:

$$M \sim Multinomial_{n+2}(N, q)$$

$$f(x_1, x_2, \cdots, x_{n+2}|N, q) = \frac{N!}{x_1! x_2! \cdots x_{n+2}!} \prod_{j=1}^{n+2} q_j^{x_j} \tag{12}$$

where $x_j$ is the sum of the column $j$ of $M$, *i.e.* the number of occurrences of genome $G_j$.

## Dirichlet prior

The Dirichlet distribution (*Dir*) is a conjugate prior for the multinomial distribution, and it has desirable mathematical properties which make it appropriate to model the composition of metagenomes.

Let $\pi_0$ indicate the prior distribution and $\pi^*$ the posterior.

From Bayes theorem,

$$\pi^*(M|\Theta) \propto L(M|\Theta)\pi_0(\Theta) \tag{13}$$

We employ a uniform prior:

$$\pi_0(\Theta) \sim Dir(\alpha)$$

$$\alpha_0 = (\alpha_1, \cdots, \alpha_{n+2}),$$

$$\alpha_i = 1 \quad \forall i = 1, \cdots, n + 2 \tag{14}$$

Because the Dirichlet is a conjugate prior for the multinomial likelihood (Eq 12), the posterior density has the same form,

*i.e*

$$\pi_*(M|\Theta) \sim Dir(\alpha^*)$$

$$\alpha^* = (\alpha_1^*, \cdots, \alpha_{n+2}^*), \tag{15}$$

$$\alpha_j^* = \alpha_j + x_j, \; j = 1, \cdots, n + 2$$

where, again, $x_j$ is the number of observations (reads) of genome $G_j$ in the assignment matrix $M$. We can then use this framework to compute the minimum abundance of a species (represented by its closest reference genome) in a metagenome.

## Posterior mean and confidence interval

Let $\gamma_j$ denote the posterior abundance of genome $G_j$. From the property of the Dirichlet we can obtain the posterior mean as follows:

$$E\left[\gamma_j|X\right] = \frac{\alpha_j^*}{\alpha^{**}}$$

$$\alpha^{**} = \sum_{k=1}^{n+2} \alpha_k^* = (n + 2) + \sum_{k=1}^{n+2} x_k = n + 2 + N \tag{16}$$

meaning that the denominator of each posterior mean corresponds to the total number of reads plus the number of categories (the number of reference genome + 2).

The marginal distribution of each abundance is:

$$\gamma_j \sim Beta(\alpha_j^*, \alpha^{**} - \alpha_j^*) \tag{17}$$

and we can use this to obtain the 95% CI numerically.

## Computational architecture

HAYSTAC is written in Python and built upon the snakemake workflow engine [47]. It operates as a wrapper around custom python scripts and several common bioinformatics tools, including: AdapterRemoval [48], bowtie2 [32], DeDup [49], mapDamage [50], samtools [51], SeqKit [52], seqtk [53] and sra-tools [54]. It uses the conda package manager, with bioconda [55], to manage its dependencies. Employing snakemake's feature to use conda for installing software dependencies required by a rule in a containerised fashion, allows HAYSTAC to have a minimum list of dependencies as a package. Furthermore, it facilitates maintaining software dependencies up to date, if necessary, without the need to make any major code changes.

## A tool for constructing databases

HAYSTAC includes a user-friendly tool to download genomes from NCBI and build customisable reference databases. After deciding *a priori* which taxa to include in their database (e.g., anywhere between a single genus to many thousands of taxa), the user constructs a database from any combination of: (i) an NCBI search query; (ii) a user specified list of NCBI accession codes; (iii) the RefSeq representative database of prokaryotic species; and/or (iv) a list of user provided reference assemblies (in FASTA format).

To construct a valid NCBI search query, visit the NCBI Nucleotide database website (https://www.ncbi.nlm.nih.gov/nucleotide/) and use the search feature to obtain a correctly formatted query string from the "Search details" box. This search query can then be used directly with HAYSTAC to automatically download and build a reference database based on the accession codes present in the resultset returned by the query.

We caution that the choice of reference database can dramatically affect species identification for all methods that use reference databases. For example, issues can arise if the reference genomes in the database are not complete, as reads may map uniquely to one reference genome because other incomplete genomes in the database do not possess that specific sequence. To mitigate this issue, HAYSTAC uses a simple heuristic to select the longest genome available per species in the input result set, which limits issues arising from unequal size of reference genomes in the database.

## Processing and aligning reads

Pre-processed reads can be directly passed onto the alignment step (Fig 1). This is useful when dealing with modern DNA. The user can also decide whether to remove sequencing adapters in the case of raw reads, and for ancient/degraded DNA collapse overlapping mate pairs using a wrapper for AdapterRemoval [48]. Reads are then aligned to a database using Bowtie2 [32]. We first align to a single Bowtie2 index that contains all the reference genomes, to obtain a BAM file that contains one single best alignment per read. For this alignment step we use the local alignment mode in Bowtie2. This allows us to discard reads that did not map to a single reference genome. This BAM is used by HAYSTAC to compute the expected $T_s/T_v$ ratio from the data, which is used as prior in Eq 4. Reads that pass this first filtering step are then

realigned to each reference genome separately using Bowtie2 (one index per reference) in end-to-end mode. We set a maximum number of mismatches to calculate the minimum alignment score during the Bowtie alignment. The maximum mismatch allowed during this second alignment step is calculated by the average read length of the input library. Allowing more mismatches results in slower run time as the number of reads included in the analytical step increases.

## Computing abundance

Alignment files are then processed to calculate a likelihood score for each read/reference alignment and build a likelihood matrix that possesses $n \times r$ entries, where $n$ is the number of reference genomes and $r$ is the number of reads that pass the first filtering stage. This is done by computing $T_s$, and $T_v$ for each alignment, and computing the likelihood using Eq 4. Here the user can change the probability of mismatch within a given alignment ($\sigma$; default 0.05) (see Eq 4). For each read we then compute a posterior probability using Eq 7. Reads are then filtered based on their posterior probability (default 0.75) and included in the Dirichlet assignment step. Reads that have not been aligned to any taxa in the database get assigned to the "Unknown Source" category, a category that absorbs reads whose reference genomes are not included in our database. Reads that have been aligned to multiple taxa in our database but obtain a posterior probability below the threshold that is set for the Dirichlet assignment are also assigned to another special category called the "Ambiguous Source", a category that includes reads that were not informative enough to be uniquely aligned to a specific taxon. Both of these categories are included in the mean posterior abundance calculations.

## Assessing evenness of coverage

We use an evenness of coverage ratio to assess if the reads assigned to a specific taxon represent a random sample of its genome or whether they are clustering around specific regions. The evenness of coverage ratio is defined as the genome coverage of a taxon over the fraction of its genome that is covered by aligned reads. In the case of a true positive identification the reads should not be clustering around specific genome regions (evenness ration< 10 as estimated from empirical datasets), something that would indicate an HGT event. This test is particularly useful for shallow sequencing data that do not allow for more than 1× genome coverage of a given reference genome.

## Representative species RefSeq database

For our analyses we build a database out of all the representative species of the prokaryotic RefSeq database. A list of all the species that were included can be found here: https://ftp.ncbi. nlm.nih.gov/genomes/GENOME_REPORTS/prok_representative_genomes.txt. In cases where the RefSeq representative database contains more than one strain for a given species, the first listed accession was picked. We also added complete genomes from the genera *Klebsiella*, *Streptococcus*, *Corynebacterium*, *Bordetella*, *Haemophilus* and *Yersinia*, for these species that were not present in the prokaryotic representative RefSeq. When possible (*i.e.*, when genome assembly with plasmid data were available) we included plasmid sequences in the database.

## Simulations

We generated 3 different simulated datasets. The first dataset was generated to determine the accuracy of different methods under simple conditions. We did not apply any chemical

damage pattern and the read length was kept constant at 60 bp. This was done by randomly sampling 10 species from the prokaryotic representative RefSeq database (see above) twice, to get two different sets of 10 species. For each set we then simulated 2 different sequencing library samples of 1 million single-end reads using gargammel [36]. One set had no human genome included and all taxa were present at 10% abundance. In the second set 25% of reads were from the human genome, and all remaining taxa were present at 7.5% abundance. This dataset included 4 simulated sample sequencing files.

The second dataset was produced to compare performance of the different tools when analysing ancient and modern DNA, and did not include the human genome. To do so, we first generated two sets of 100 species randomly sampled from the prokaryotic representative RefSeq database, and two sets of 500 randomly sampled species. The species were different between each of the 2 random samplings for both the 100 species and 500 species sets. For each species profile in both the 100 species and 500 species sets, we simulated two different libraries (8 total libraries) using gargammel with 10 million single-end reads of length 125bp from an Illumina HiSeq 2500 run. The species abundances were kept consistent within each library, where libraries of 100 species have all taxa at 1% abundance and libraries of 500 species have all taxa at 0.2% abundance. The first library incorporated aDNA damage patterns based on the profile of reads that mapped to the *Tannerella forsythia* genome from a real ancient dental calculus sample (CS21 from [43]) and the second library did not include aDNA damage patterns.

To assess our method with more realistic data sets, we analysed simulated data previously published by Velsko et al. [30] designed to resemble a dental plaque community, which included species that were not in the prokaryotic representative RefSeq database. Twelve datasets of 200 species each were generated: with and without ancient DNA damage, each with even species abundance (0.5%) or with abundance based on values observed in an ancient oral microbiome bacterial community (see S8 Table). The type of simulation included members of the *Corynebacterium* genus without *Corynebacterium diphtheriae*; the second type included all members of the *Corynebacterium* genus (including *Corynebacterium diphtheriae*); the third type did not include any member of the *Corynebacterium* genus. Each of these twelve samples contained 5 million paired end reads of varying length, that were collapsed with AdapterRemoval.

Lastly, we simulated reads from human pathogens and closely related species to test HAYSTAC's ability to identify low abundance pathogens in metagenomic data. To do so, we simulated 200 single-end Illumina sequencing libraries from 100 pathobionts and 100 non-pathogenic closely related species respectively, with aDNA damage patterns (estimated from simulated sample anc200e2repgn), which we then added to one of the simulated ancient Oral Microbiome dataset samples (anc200e2repgn). These libraries were then analysed using both HAYSTAC and HOPS (see below for more details).

## Analyses of simulated data

We trimmed adapters using AdapterRemoval v2 [48] excluding reads that were smaller than 15 bp. Analyses of simulated data with Kraken2/Bracken [23, 27], KrakenUniq [24], MALT [25] were conducted using default parameters. HAYSTAC and Sigma [3] were run with default parameters. For HAYSTAC, Kraken2/Bracken, KrakenUniq, MALT and Sigma positive identification abundance threshold was set to 0.01% while we used 50 reads for MALT/HOPS as in Hübler et al. [28].

For the pathogen detection analysis, HAYSTAC was run with a mismatch probability of 0.2 and a read assignment probability threshold of 0.5. For the HOPS analysis we initially ran

cMALT (a version of MALT adapted to aDNA libraries) with the following flags activated $-asm-tails5$, while setting all the other parameters at default settings. The cMALT output was subsequently analysed with HOPS [28] with default values but providing a list of candidate species that included 100 pathogen species, and using the outputs characterised as ancient by HOPS. Before counting the true and false positive reads assigned to pathogen taxa by each method, we removed the all the microbial taxa each method would identify in the initial library of the ancient Oral Microbiome sample anc200e2repgn, so that the background species composition of the library would not affect the identification of the spiked pathobionts.

In order to assess how HAYSTAC, Sigma, Kraken2/Bracken, KrakenUniq and MALT perform with genus specific database, we analysed 20 simulated samples (based on RefSeq genomes; see above) and aligning reads to all representative genomes in a single genus. Nine genera were picked randomly for these analyses: *Bacteroides* (25 species), *Burkholderia* (3 species), *Campylobacter* (23 species), *Clostridium* (91 species), *Corynebacterium* (84 species), *Desulfitobacterium* (4 species), *Mycobacterium* (36 species), *Solimonas* (3 species) and *Streptococcus* (78 species).

## Case study 1: Ancient *Yersinia pestis*

We tested HAYSTAC using a published dataset of ancient human bone and tooth samples in which *Yersinia pestis* was identified [9], to compare performance against standard techniques used to identify specific pathogens in aDNA samples. We downloaded reads from all seven samples in which the authors detected *Y. pestis* (RISE00, RISE139, RISE386, RISE505, RISE509, RISE511) from the European Nucleotide Archive (NCBI BioProject accession: PRJEB10885). We then used HAYSTAC to build a database that contains the longest complete genome for each species of the genus *Yersinia* and aligned reads using default parameters. HAYSTAC was then used to compute posterior abundance of each species.

## Case study 2: Historic dental calculus

We also tested HAYSTAC using an historic dental calculus dataset [43] (PRJEB30331). Raw reads were processed as described in Velsko et al. [43]. We attempted to identify a set of respiratory pathogens in these calculus samples, including *Corynebacterium diphtheriae* and *Bordetella pertussis*, the causative agents of diphtheria and whooping cough, respectively (Fig 8). We also tested for the presence of species belonging to the *Haemophilus* (ideally *Heamophilus influenzae* or *parainfluenzae*), *Klebsiella* (ideally *Klebsiella pneumoniae*) and *Streptococcus* (*Streptococcus pneumoniae*) genera. These species were selected to test if pathobionts of the upper respiratory system could be found in dental calculus.

## Performance test

For the performance tests we created a reproducible conda environment, inside which all the different pipelines were installed and run. For each method we used the GNU time command to measure elapsed time (wall clock) and peak memory usage (maximum resident set size). Scripts to run the performance benchmarks are available from https://github.com/antonisdim/haystac_paper

We used a total of 500 accessions, one for each species, which were a subset of the accessions used for the analysis of all real and simulated datasets. We built databases with an increasing number of species (10, 100, 500) and the same database was given as input to all the methods. We only measured the database building time and maximum memory usage for each method, as HAYSTAC was used to efficiently fetch all 500 reference genomes from NCBI.

For sample inputs we used samples from the Human Microbiome Project, with accession SRS078671. We merged the forward-mate and the singleton reads in one file that we then treated as single end reads. That file was subsequently subsampled to generate fastq files of 10 K, 100 K, and 1 M SE reads respectively, that were used as inputs for all the tests.

For the sample analysis we performed two tests, one that measured max memory usage and runtime for samples of increasing size against the database of 500 species, and a second where the input file size was kept constant at 1 million reads, while the database size was being varied, again measuring the same variables for memory and elapsed execution time.

## Supporting information

**S1 Appendix. Calculation of false positive, false negative and true positive rates.**
(PDF)

**S1 Fig. Benchmarking for elapsed runtime and memory for HAYSTAC, Sigma, Kraken2/ Bracken, KrakenUniq and MALT when analysing samples of 10 K, 100 K and 1 M reads against a database of 500 species.** Memory remains constant as sample size increases, and runtime in most methods (other than Kraken2/Bracken and KrakenUniq) scales with the database size rather than input sample size).
(PDF)

**S2 Fig. False positive and negative rates per method for the simulated samples of the General Microbiome dataset.**
(PDF)

**S3 Fig.** Mean count of false positive (red), false negative (orange), and true detected species (blue) in the simulated General Microbiome dataset of 100 species ancient (n = 2) (A), 100 species modern (n = 2) (B), and 500 species ancient (n = 2) (C) and 500 species modern (n = 2) (D). The dotted line represents the average number of simulated species in each set of samples, and the numbers above the error bars represent the mean species count.
(PDF)

**S4 Fig. False positive and negative rates per method for the simulated samples of the Oral Microbiome dataset.**
(PDF)

**S5 Fig. Receiver operator curve analysis, showing the relationship between the true and false positive ratios (TPR and FPR respectively) to determine the default read posterior probability threshold for the Dirichlet assignment across all simulated samples.** A rather conservative threshold of 0.75 is appropriate for most types of analysis. If the user requires a more relaxed threshold for discovery purposes the threshold can be lowered to 0.5. Respectively if a user is analysing deep sequencing data or wants to perform a more stringent identification the threshold can also be increased.
(PDF)

**S6 Fig. Receiver operator curve analysis, showing the relationship between the true and false positive ratios (TPR and FPR respectively) to determine the default read posterior probability threshold for the Dirichlet assignment per sample.** Here we can see how the read length and deamination levels affect the true positive rate. The user might want to consider these additional factors if they wish to change the default value of 0.75.
(PDF)

**S1 Table. Characteristics of the General Microbiome simulated dataset.**
(XLSX)

**S2 Table. Characteristics of the Oral Microbiome simulated dataset.**
(XLSX)

**S3 Table. Species identification counts and rates for the General Microbiome simulated dataset.**
(XLSX)

**S4 Table. Species identification counts and rates for the Oral Microbiome simulated dataset.**
(XLSX)

**S5 Table. Mean species counts per genus per method for the genus database analysis.**
(XLSX)

**S6 Table. Oral Microbiome dataset simulated species.** A total of 196 species was simulated for the Oral Microbiome dataset. Only 115 species were included in the database, with only 4 of these species sharing a common accession.
(XLSX)

**S7 Table. False positive species, whose RefSeq reference genomes contain human contamination.**
(XLSX)

**S8 Table. Table of species simulated for the Oral Microbiome dataset.** Samples containing an 'l' character in their names contain species in various abundances, whereas samples with an 'e' in their name contain species in constant abundances throughout. Genus negative control samples have the whole *Corynebacterium* genus removed, whereas species negative controls do not include *Corynebacterium diphtheriae*.
(XLSX)

**S9 Table. Memory and time usage data for database construction across HAYSTAC, Kraken2/Bracken, KrakenUniq, MALT and SIGMA.**
(XLSX)

**S10 Table. Memory and time usage data for sample analysis across HAYSTAC, Kraken2/Bracken, KrakenUniq, MALT and SIGMA.**
(XLSX)

**S11 Table. Pathogenic and non-pathogenic species used for the pathogen detection analysis.**
(XLSX)

## Acknowledgments

The authors would like to acknowledge the use of the University of Oxford Advanced Research Computing (ARC), (http://dx.doi.org/10.5281/zenodo.22558), Queen Mary's Apocrita HPC facility and the HPC facilities of the Leibniz Supercomputing Centre. The authors would also like to thank Felix Key for his helpful feedback on the comparison with HOPS, and the design of the pathogen identification experiment, as well as Antonio Fernandez Guerra for his feedback during the early stages of the manuscript writing process, beta testing of the HAYSTAC package and insight into the biological interpretation of the results.

## Author Contributions

**Conceptualization:** Evangelos A. Dimopoulos, Alberto Carmagnini, Irina M. Velsko, Christina Warinner, Greger Larson, Laurent A. F. Frantz, Evan K. Irving-Pease.

**Data curation:** Evangelos A. Dimopoulos, Irina M. Velsko, Evan K. Irving-Pease.

**Formal analysis:** Evangelos A. Dimopoulos, Alberto Carmagnini, Irina M. Velsko, Evan K. Irving-Pease.

**Funding acquisition:** Greger Larson, Laurent A. F. Frantz.

**Methodology:** Evangelos A. Dimopoulos, Alberto Carmagnini, Christina Warinner, Laurent A. F. Frantz, Evan K. Irving-Pease.

**Project administration:** Christina Warinner, Greger Larson, Laurent A. F. Frantz, Evan K. Irving-Pease.

**Resources:** Christina Warinner, Greger Larson, Laurent A. F. Frantz, Evan K. Irving-Pease.

**Software:** Evangelos A. Dimopoulos, Evan K. Irving-Pease.

**Supervision:** Irina M. Velsko, Christina Warinner, Greger Larson, Laurent A. F. Frantz, Evan K. Irving-Pease.

**Validation:** Evangelos A. Dimopoulos, Alberto Carmagnini, Evan K. Irving-Pease.

**Visualization:** Evangelos A. Dimopoulos, Evan K. Irving-Pease.

**Writing – original draft:** Evangelos A. Dimopoulos, Alberto Carmagnini, Laurent A. F. Frantz, Evan K. Irving-Pease.

**Writing – review & editing:** Irina M. Velsko, Christina Warinner, Greger Larson, Laurent A. F. Frantz, Evan K. Irving-Pease.

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
