## [Decision Letter · Decision Letter 0]

8 Apr 2022

Dear Dr Irving-Pease,

Thank you very much for submitting your manuscript "HAYSTAC: A Bayesian framework for robust and rapid species identification in high-throughput sequencing data" for consideration at PLOS Computational Biology.

As with all papers reviewed by the journal, your manuscript was reviewed by members of the editorial board and by several independent reviewers. In light of the reviews (below this email), we would like to invite the resubmission of a revised version that takes into account the reviewers' comments.

We cannot make any decision about publication until we have seen the revised manuscript and your response to the reviewers' comments. Your revised manuscript is also likely to be sent to reviewers for further evaluation.

Sincerely,

Niranjan Nagarajan

Associate Editor

PLOS Computational Biology

William Noble

Deputy Editor

PLOS Computational Biology

Reviewer's Responses to Questions

**Comments to the Authors:**

Reviewer #1: This is a very interesting and well written paper, that describes a nice approach to identifying species in metagenomic sequencing data from both current and ancient DNA.

It starts by doing a two-tier sequence alignment calculation, first using bowtie to determine which reads have alignments and then aligning each read to each reference genome separately, this addressing the issue that bowtie is a best-mapper and not an all-mapper (unlike e.g. MALT).

Transitions and transversions are counted to estimate similarity to reference genomes.

A Bayesian framework is used to estimate the probability of a species being present in a sample, also taking evenness of coverage into account.

The program is shown to compare favorably with other widely-used programs that address the same questions. This is based on simulated data. The program is also run on real datasets and the authors show that additional insights may be obtained compared to the original analyses of the datasets.

I am happy with most of the manuscript, however one topic I feel is not correctly addressed.

In lines 145 - 160 the authors write that HAYSTAC runs in "a single step".

First, this is a silly thing to emphasize given that Kraken/Braken and MALT/HOPS both run in only two steps and that anyone processing more than one sample will obviously, and easily, write those two steps back-to-back in a script.

Second, more seriously, this distracts from the issue that HAYSTAC is not a single program, but rather a Python script that uses Snakemake to call eight different programs. This is a brittle construct.

While the use of bioconda and Snakemake ensure that all necessary versions of the programs are put in place when the tool is used, my worry would be that the HAYSTAC is "frozen in time" and moving forward it will be difficult to benefit from updates to any of the eight ingredient programs.

In constrast, the Kraken2/Braken pipeline consists of two programs, and the described MALT/HOPS pipeline consist of two self-contained Java programs. In both cases, this is a much less brittle architecture.

I think that you should be forthright about the fact that HAYSTAC has many more moving parts than the other tools and discuss whether this is problematic or not.

Reviewer #2: In this paper, the authors present an approach based on a Bayesian framework to estimate the abundance and statistical support for species identification. Unlike previous methods, HAYSTAC is intended to handle not only ancient DNA data but also modern DNA data and incomplete reference databases.

This approach is interesting and pertains to the niche of ancient organism identification in particular. The source code and some data are available for download. However, I would like to suggest some improvements.

Major

1) What is the rationale for choosing Kraken2/Bracken instead of KrakenUniq?

It has been demonstrated in Ye,2019 (Benchmarking Metagenomics Tools for Taxonomic Classification) that KrakenUniq is less prone to false-positive results than Kraken2. If such a baseline could be included, the analysis and claim reported in Figures 3 and 4 would be significantly enriched.

2) The author claims to have developed a "rapid species identification" tool. According to my understanding, lines 516 to 520 support this claim. This relates to the use of reduced reference databases for genera, which reduces the computational time (350 times faster). As shown in figure 2, Kraken2 provides a faster run time for analysis. Including KrakenUniq in Figure 5 ("Genus specific database analysis") would enhance your claims, especially in reporting the computational time and accuracy over KrakenUnique in this case.

Minor

1) Please provide a brief explanation of why Sigma presented fewer false positives for modern DNA than Haystack in Table 2.

2) The code and data used to generate the figures in the paper are not provided. Where it is possible, such data/code should be made available in order to enhance reproducibility in the paper.

**Have the authors made all data and (if applicable) computational code underlying the findings in their manuscript fully available?**

Reviewer #1: Yes

Reviewer #2: Yes

PLOS authors have the option to publish the peer review history of their article (what does this mean?). If published, this will include your full peer review and any attached files.

Reviewer #1: No

Reviewer #2: No
---

## [Decision Letter · Decision Letter 1]

16 Aug 2022

Dear Dr Irving-Pease,

We are pleased to inform you that your manuscript 'HAYSTAC: A Bayesian framework for robust and rapid species identification in high-throughput sequencing data' has been provisionally accepted for publication in PLOS Computational Biology.

Best regards,

Niranjan Nagarajan

Academic Editor

PLOS Computational Biology

William Noble

Section Editor

PLOS Computational Biology

Reviewer's Responses to Questions

**Comments to the Authors:**

Reviewer #2: Dear authors, thank you for taking the time and effort to improve this work as a result of our discussion.

Thus, my concerns have been addressed by the authors, and I recommend acceptance of this manuscript.

**Have the authors made all data and (if applicable) computational code underlying the findings in their manuscript fully available?**

Reviewer #2: Yes

PLOS authors have the option to publish the peer review history of their article (what does this mean?). If published, this will include your full peer review and any attached files.

Reviewer #2: No

---

## [Editor Report · Acceptance letter]

26 Sep 2022

PCOMPBIOL-D-22-00306R1 

HAYSTAC: A Bayesian framework for robust and rapid species identification in high-throughput sequencing data

Dear Dr Irving-Pease,

I am pleased to inform you that your manuscript has been formally accepted for publication in PLOS Computational Biology. Your manuscript is now with our production department and you will be notified of the publication date in due course.

With kind regards,

Zsofi Zombor
